TOPICAL REVIEW

# Blue plaque review series: A.V. Hill, athletic records and the birth of exercise physiology

Mark Burnley[1] (iD), Anni Vanhatalo[2] (iD), David C. Poole[3] (iD) and Andrew M. Jones[2] (iD)

[1]*School of Sport, Exercise and Health Sciences, Loughborough University, Loughborough, UK*
[2]*Public Health and Sport Sciences, University of Exeter, Exeter, UK*
[3]*Departments of Kinesiology and Anatomy and Physiology, Kansas State University, Kansas, USA*

Handling Editors: Laura Bennet & Bruno Grassi

The peer review history is available in the Supporting Information section of this article (https://doi.org/10.1113/JP288130#support-information-section).

**Abstract figure legend** Depiction of the world running records from 1925 studied by A. V. Hill and those of 2025 up to and including ultradistance events on a semi-logarithmic plot. These records, in turn, have a clear bioenergetic basis [as shown in the plot of oxygen uptake against time, with blue circles showing exercise performed below 'critical power' where a steady state or 'dynamic equilibrium' can be attained, and the red circles show exercise above critical power, where there is no steady state (a loss of dynamic equilibrium)]. Despite preceding the discovery of ATP by 4 years, Hill's bioenergetic interpretations were remarkably close to the currently accepted view. Hill discussed several other concepts in his papers, including carbohydrate supplementation, pacing strategy, sex differences, comparative physiology and high jump mechanics. Many of his ideas on these matters were decades ahead of their time.

**Abstract** One hundred years ago, A.V. Hill authored three manuscripts analysing athletic world records from a physiological perspective. That analysis, grounded in Hill's understanding of contemporary muscle bioenergetics, provides a fascinating sketch of the thoughts and speculations of one of the fathers of exercise physiology. In this review, we reflect on Hill's prose with the benefit of 100 years of hindsight, and illustrate how Hill was able to draw startlingly accurate conclusions from what limited data were available on the physiology of intense exercise. Hill discusses the energetics of running, swimming, rowing and cycling in both males and females, as well as addressing exercise performance in horses and the mechanics of jumping. He also considers sports nutrition, pacing strategy and ultra-endurance exercise. Perhaps most impactfully, he establishes that the speed–duration relationship has characteristics that reflect the underlying physiological basis of exercise performance. That physiology, in turn, differs depending on the duration of the event itself, providing one of the first descriptions of the task-dependent nature of mechanisms limiting exercise tolerance. A remarkable feature of Hill's papers is that they were written just a few years before a major revolution in muscle biochemistry, and yet Hill was still able to develop conceptually sound ideas about human performance. His hypotheses require only minor revision to bring them into line with current understanding. In reaching their centenary, therefore, the surprising feature of these papers is not how well they have aged, but how relevant they remain.

(Received 5 December 2024; accepted after revision 6 February 2025; first published online 22 February 2025)

**Corresponding author** M. Burnley: School of Sport, Exercise and Health Sciences, Loughborough University, Loughborough, Leicestershire, UK. Email: M.Burnley@lboro.ac.uk

## Introduction

The year 2025 marks the centenary of A.V. Hill's papers on the physiological basis of athletic records, published in *The Lancet* (Hill, 1925a), *Nature* (Hill, 1925b) and *The Scientific Monthly* (Hill, 1925c). The source material for these papers came from a lecture Hill delivered to the physiology section of the British Association for the Advancement of Science in September 1925. The papers are almost identical, with the *Lancet* and *Nature* papers being edited versions missing several contextual passages from *The Scientific Monthly*. The *Lancet* paper (Hill, 1925a) includes an editor's note omitting part of the work delivered by 'the lecturer', suggesting that all three journals offered or agreed to publish the lecture transcript in some form. Their contents provide a fascinating view of exercise physiology in its infancy, as well as of the thoughts of the 1922 Nobel Laureate who helped bring the discipline into being.

The papers were written at a time when the understanding of muscle biochemistry was rudimentary, but the inferences Hill made in relation to the bioenergetics of exercise performance are remarkably similar, conceptually at least, to current understanding. Hill also made a series of observations regarding exercise intensity, exercise modality, female athletic performance, sports nutrition and comparative physiology, many of which remain pertinent today. The aim of this review is to examine Hill's 1925 papers in detail, to consider their place in the history of physiology and to place Hill's observations in light of

**Mark Burnley** is a Senior Lecturer in exercise physiology at Loughborough University. Anni Vanhatalo is Professor of Human Physiology at the University of Exeter. David C. Poole is University Distinguished Professor of Kinesiology and Physiology at Kansas State University. Andrew M. Jones is Professor of Applied Physiology at the University of Exeter. Collectively, this group investigates the integrative physiology of exercise tolerance using a range of experimental models and techniques. This has included studies on the mechanistic basis of the power–duration relationship, the origin of the slow component of oxygen uptake kinetics, microvascular function, the nitric oxide-nitrate-nitrite pathway and its influence on the exercise response, and the neuromuscular and metabolic mechanisms of muscle fatigue.

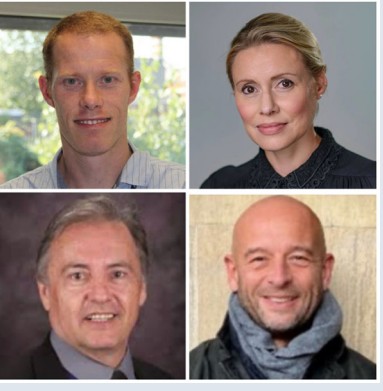

what we know 100 years later. Given the extra detail presented within, we will refer to *The Scientific Monthly* paper (Hill, 1925c) unless otherwise stated.

## Curve fitting and its interpretation

Hill was attempting to find answers to the questions 'How fast can athletes run a given distance?' and 'What are the factors that determine the variance in speed with distance?' Hill initially addressed them by presenting the world records of the time for running, swimming, cycling and rowing (Fig. 1). Notably, the paper does not provide equations associated with any of the curves describing the speed–duration relationships, which would have been hand drawn, with the error about the fitted line determined by eye. Figure construction in the pre-computer age was a labour-intensive process requiring a high degree of technical skill, and Hill was a master of the art. Least squares non-linear regression algorithms commonly used today were not developed until the mid-1940s (Levenberg, 1944; Marquardt, 1963). Furthermore, the Levenberg–Marquardt algorithm did not become widely available in statistical and graphing software for personal computers until the 1990s. In lieu of equations and the algorithm to fit them, Hill described the plots in Fig. 1 thus:

'It is obvious in all four cases that we are dealing with the same phenomena, a very high speed maintainable for short times, a speed rapidly decreasing as the time is increased and attaining practically a constant value after about 12 min.' (Hill, 1925c; p. 421).

This interpretation implies that the speed–duration relationship is asymptotic, in common with both exponential (Bundle & Weyand, 2012; Rohmert, 1960) and hyperbolic (Monod & Sherrer, 1965; Moritani et al., 1981; Poole et al., 1988) formulations used subsequently. However, an alternative description of the speed–duration relationship is that it conforms to a power law (Drake et al., 2024; Katz & Katz, 1994, 1999; Kennelly, 1906), in which asymptotic behaviour is absent. Notably, Kennelly (1906) is one of the papers that Hill (1925c) cites in this context:

'I will not deal further with the statistical analysis of the facts, beyond referring to an extremely interesting and suggestive collection of them given in a paper by A. E. Kennelly … Some, indeed, of my data are taken directly from that paper.' (Hill, 1925c; p. 409).

Hill made no statistical attempt to establish the 'best' curve to fit to the data. We have applied all three of the above models to Hill's data in our Fig. 2:

**Hyperbolic model**:

$$t = D' / (S - CS) \qquad (1)$$

where $t$ is race time (s), $D'$ ('distance prime') is the curvature constant parameter (m), $S$ is speed and $CS$ is critical speed (both in m s$^{-1}$) (Monod & Scherrer,

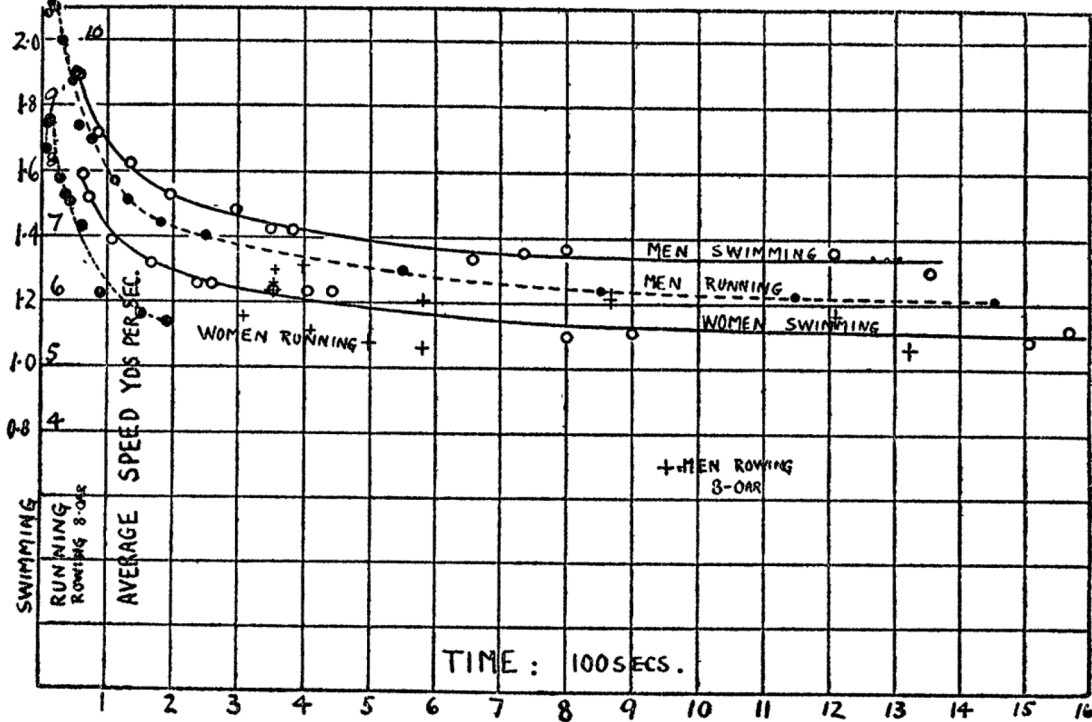

**Figure 1. AV Hill's plot of world records**
AV Hill's plot of world records in 1925 for running, swimming and rowing. Reproduced from Hill (1925c).

1965). Wilkie (1960, 1980) developed a hyperbolic power–duration model for cycling based on the concepts of maximal aerobic power, anaerobic capacity and oxygen uptake ($\dot{V}_{O_2}$) kinetics (for additional details, see Morton & Hodgson, 1996):

$$S = E + A/t - E \times \tau \times (1 - e^{-t/\tau})/t \qquad (2)$$

where $A$ is the anaerobic capacity, $E$ is the 'maximal aerobic power' and $\tau$ (fixed at 10 s) is equivalent to the time constant of the increase in the aerobic contribution to the speed–duration curve at time $t$, loosely related to the kinetics of $\dot{V}_{O_2}$ at the onset of exercise. Note that this model is a variant of eqn (1) solved for speed (i.e. $S = D'/t + CS$) and the exponential term effectively increases the rate at which the speed–duration curve initially declines because the value $E$, equivalent to the critical speed, is not fully expressed for several minutes. Wilkie (1980) formulated his equation independently of Monod and Scherrer (1965). Wilkie's model is not included in Fig. 2.

**Exponential model**:

$$S = S_{\text{aer}} + (S_{\text{max}} - S_{\text{aer}}) \times e^{-kt} \qquad (3)$$

where $S$ is speed, $S_{\text{aer}}$ and $S_{\text{max}}$ are aerobic speed and maximal speed, respectively (m s$^{-1}$), $t$ is time (s) and $k$ is the rate constant (Bundle & Weyand, 2012).

**Power law model**:

$$S = a \cdot t^b \qquad (4)$$

where $S$ is speed (m s$^{-1}$), $t$ is time (s), $a$ is the intercept, notionally equivalent to maximal instantaneous sprinting speed and $b$ is the scaling exponent relating the decay in speed to time (Kennelly, 1906).

Interestingly, none of these models replicate the curve fit provided by Hill (1925c): the hyperbolic model adequately characterises the longer records and projects towards an asymptote, but the model error associated with events <2 min in duration is substantial; the exponential model provides a good approximation of short-distance records and predicts asymptotic behaviour, but substantial deviations can be seen at middle- and long distances; lastly, the power law model provides a good general approximation of the speed–duration curve, but shows significant errors for middle-distance records and for the longest race in particular.

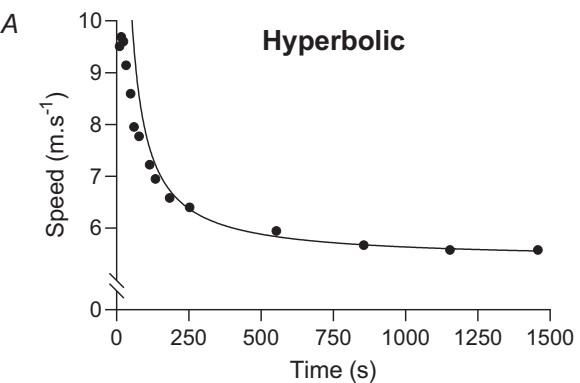

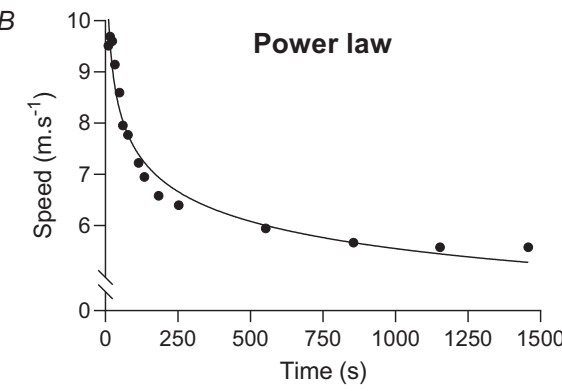

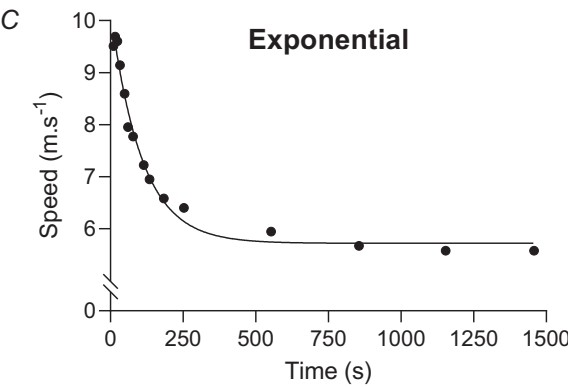

**Figure 2. Male running world records**
Male running world records in 1925 redrawn from Hill (1925c), with hyperbolic (*A*), power law (*B*) and exponential (*C*) functions. Data were plotted following extraction using webplotdigitizer (https://apps.automeris.io/wpd), and curves were fit using non-linear regression in Prism, version 9 (GraphPad Software Inc., San Diego, CA, USA). For further details, see text.

## Bioenergetics and the concept of a 'critical speed'

Hill's speed–duration curves are a composite of the performance of several different athletes participating in races in different locations under unknown environmental conditions. They do not, therefore, represent the speed–duration relationship in an individual participant exercising to task failure under strictly controlled conditions in which similar baseline physiological status exists (i.e. fitness and environmental conditions do not change across testing bouts). Moreover, Hill's curves are not, in and of themselves, appropriate for building or testing theory when robust studies can achieve that purpose. That said, Hill's interpretation of the curves in question was grounded in his understanding of exercise bioenergetics at the time:

'The maximum effort, therefore, which [an athlete] can exert over a given interval depends upon the amount of energy available for him, upon (1) his maximum oxygen intake (that is, his income), and (2) his maximum oxygen debt (that is, the degree to which he is able to overdraw his account).' (Hill, 1925c; pp 412–413).

The contemporary understanding of high-intensity exercise tolerance is similar to this interpretation, insofar as exercise tolerance during severe-intensity exercise (i.e. above the critical speed, the domain in which many Olympic athletic events take place) is determined, in part, by the interaction of maximal oxygen uptake ($\dot{V}_{O_2max}$) and the capacity for substrate-level phosphorylation (Burnley & Jones, 2007). However, modern theory deviates somewhat from Hill's conceptual understanding in two key respects. First, the tolerable duration of exercise was considered to be the consequence of the accumulation of a maximal $O_2$ deficit, whereas it is now considered to be a function of both the depletion of high-energy phosphates and the accumulation of fatiguing metabolites (particularly inorganic phosphate and $H^+$)

(Cannon et al., 2013; Jones, Wilkerson, DiMenna et al., 2008; Korzeniewski & Rossiter, 2021). These changes, in turn, have neurophysiological and perceptual effects which may also limit exercise tolerance (Hureau et al., 2018). Second, in earlier works Hill interpreted the transition from steady state to non-steady state as occurring at the running speed associated with $\dot{V}_{O_2max}$ (Hill & Lupton, 1923) (Fig. 3), even using the phrase 'critical speed' to describe it:

'Considering the case of running, there is clearly some critical speed for each individual, below which there is a genuine dynamic equilibrium, break-down being balanced by restoration, above which, however, the maximum oxygen intake is inadequate, lactic acid accumulating, a continuously increasing oxygen debt being incurred, fatigue, and exhaustion setting in.' (Hill & Lupton, 1923; p. 151).

By contrast, the modern understanding of these responses is that non-steady state behaviour emerges at a critical $\dot{V}_{O_2}$ value (and critical running speed) substantially below $\dot{V}_{O_2max}$ (Goulding & Marwood, 2023;

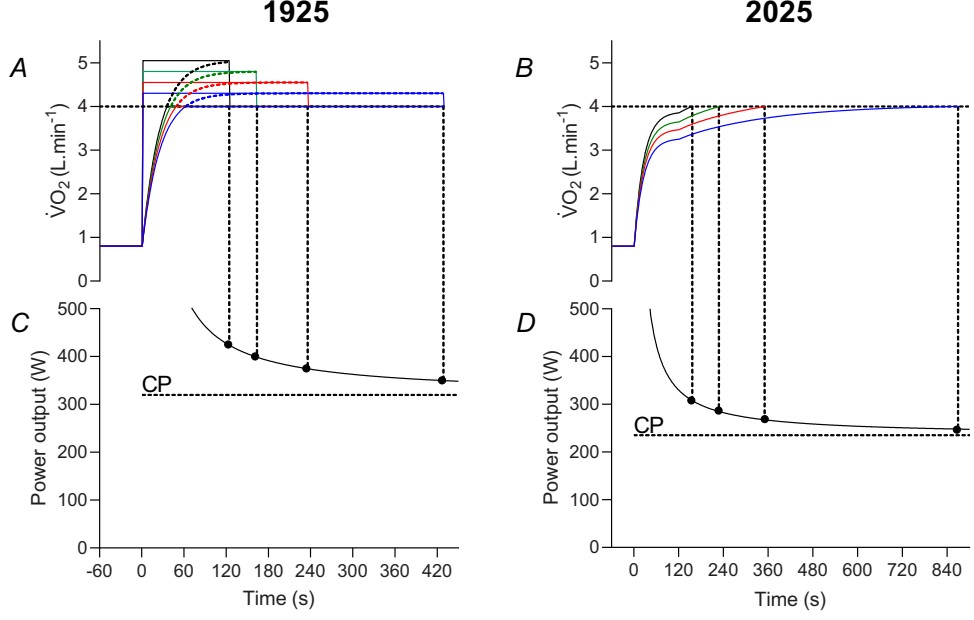

**Figure 3. Oxygen and power–duration during severe-intensity cycle ergometry**
Schematic illustrations of the oxygen uptake and power–duration relationships during severe-intensity cycle ergometry, showing (*A* and *C*) interpretation of Hill (1925c) and (*B* and *D*) those of contemporary measurements (Burnley & Jones, 2007; Murgatroyd et al., 2011). In (*A*), it is assumed that non-steady state behaviour emerges when $\dot{V}_{O_2max}$ is exceeded (tasks at 350 (blue), 375 (red), 400 (green) and 425 W (black) are shown). Dashed lines for $\dot{V}_{O_2}$ above $\dot{V}_{O_2max}$ indicate the projected rather than actual $\dot{V}_{O_2}$. Exercise tolerance is then dictated by the maximal accumulated $O_2$ deficit in each exercise bout (represented by the coloured rectangles above $\dot{V}_{O_2max}$, in addition to the $O_2$ deficit accumulated before the attainment of $\dot{V}_{O_2max}$). The resulting power–duration relationship is hyperbolic, with a curvature constant (*W'*) of 13.0 kJ and a critical power occurring at the power output associated with V·$O_2$max during incremental exercise (320 W) (*C*). In (*B*), the $\dot{V}_{O_2}$ responses include a slow component which results in the attainment of $\dot{V}_{O_2max}$ and exercise intolerance [tasks at 247 (blue), 269 (red), 287 (green) and 308 W (black) are shown]. The power–duration relationship is also hyperbolic, with a W' of 11.4 kJ. The critical power now occurs at a lower power output (235 W) (*D*). In both plots, it is assumed that the primary $\dot{V}_{O_2}$ gain is 10 mL min$^{-1}$ W$^{-1}$ (equivalent to a mechanical efficiency of ~29%) and the primary (phase II) time constant is 25 s. The $\dot{V}_{O_2}$ slow component has a 120 s time delay, a 250 s time constant and amplitude of 0.75–0.90 L min$^{-1}$.

Jones et al., 2010; Poole et al., 2016). At such speeds, in the severe intensity domain, $\dot{V}_{O_2}$ increases progressively to attain $\dot{V}_{O_2max}$ because of the development of the slow component of $\dot{V}_{O_2}$ kinetics (Poole et al, 1988; Whipp & Mahler, 1980; Whipp, 1994). At the same time, muscle phosphocreatine (PCr) progressively falls, and inorganic phosphate progressively rises, indicating an obligatory energetic contribution from substrate-level phosphorylation (Jones, Wilkerson, DiMenna et al., 2008; Vanhatalo et al., 2010, 2016). This loss of what Hill termed 'dynamic equilibrium' at speeds below $\dot{V}_{O_2max}$ but above the critical speed thus initiates a chain of events that ultimately limits exercise tolerance. It is important to note that 'dynamic equilibrium' is a more thermo-dynamically precise description than 'steady state' because metabolically relevant variables, notably muscle glycogen (Black et al., 2017), continue to change under conditions of a steady state $\dot{V}_{O_2}$ (below the critical speed).

Above the critical speed, the trajectory of the $\dot{V}_{O_2}$ slow component determines when $\dot{V}_{O_2max}$ is reached, with task failure occurring soon thereafter (Burnley & Jones, 2007; Murgatroyd et al., 2011; Whipp & Ward, 1992a). Consequently, it is the interaction between the trajectory of the $\dot{V}_{O_2}$ slow component, $\dot{V}_{O_2max}$ and the capacity to derive energy from substrate-level phosphorylation (PCr hydrolysis, glycolysis leading to the formation of lactate and the adenylate kinase reaction, Hill's 'oxygen debt') that dictates the tolerable duration of exercise in the severe-intensity domain (Burnley & Jones, 2007; Murgatroyd et al., 2011; Whipp, 1994). Those inter-actions, in turn, shape the non-linear relationship between running speed and exercise duration: without the $\dot{V}_{O_2}$ slow component, the critical speed asymptote would be determined by the accumulation of the maximal $O_2$ deficit, and, as a consequence, would occur at the speed associated with $\dot{V}_{O_2max}$ (Fig. 3*C*). The non-steady state behaviour of the $\dot{V}_{O_2}$ slow component means that the critical speed asymptote necessarily occurs at a running speed below $\dot{V}_{O_2max}$ (Fig. 3*D*).

Although Hill's conceptual understanding of exercise bioenergetics was correct, the details were not, with the focus on lactic acid as a control factor in the $\dot{V}_{O_2}$ response and the idea that the maximal $O_2$ debt was ∼15 L being two obvious examples. In the former case, PCr and ATP had not been discovered at the time of Hill's lecture, and the non-oxidative energy contribution is now calculated using the within-exercise $O_2$ deficit rather than the post-exercise $O_2$ debt (maximal $O_2$ deficits of ∼3–6 L during running are typical) (Medbø et al., 1988), although it should be noted that due to non-linearities in the relationship between work rate and muscle energetics the maximal $O_2$ deficit calculation itself is problematic (Bearden & Moffatt, 2000; Gaesser & Poole, 1996; Özyener et al., 2003; Whipp, 1994; Wilkerson et al., 2004). Viewed differently, however, these examples illustrate

Hill's uncanny ability to derive essentially the right answer from experimental data: the $\dot{V}_{O_2}$ response *is* controlled by changes in intramuscular substrates (Rossiter et al., 1999, 2002; Whipp & Mahler, 1980) and there *is* a finite capacity for substrate-level phosphorylation, albeit one that is difficult to quantify (Medbø et al., 1988; Noordhof et al., 2013).

## The revolution in muscle physiology

The pace with which the understanding of bioenergetics developed following Hill's papers was astonishing: in 1927 Grace and Philip Eggleton, working in the same department as Hill in University College London, would isolate and identify a 'labile phosphorus' (Eggleton & Eggleton, 1927), with Fiske and Subbarow (1927) independently showing that this compound was bound to creatine, naming it 'phosphocreatine'. Just 2 years later, Lohmann published his discovery of ATP (Lohmann, 1929). Fiske and Subbarow (1929) independently confirmed its existence soon afterwards. Hill justifiably considered these works a 'revolution in muscle physio-logy' (Hill, 1932). His enthusiastic acceptance of the new high-energy phosphate mechanism of bioenergetics played a key role in overturning theories he himself developed:

'I am ready, as you will see, to bear my share of the blame for an imperfect theory; as a matter of fact, however, so long as oxidative recovery follows activity, most of these applications to man are unaffected. It matters little to the oxygen debt after a bout of exercise whether we attribute it to 'lactic acid formation' alone, or add 'and phosphagen breakdown'. It makes no difference to the mechanical efficiency whether the energy for contraction is derived chiefly from phosphagen, or chiefly from lactic acid. This is because the new facts in man were discovered by experiments conceived on the basis of the older facts in isolated muscle: they do not depend on any particular theory.' (Hill, 1932).

By the mid-1930s, several lines of evidence placed ATP as the probable primary energy donor during muscle contraction; for a comprehensive historical review, see Rall (2023). Hill (1950) challenged biochemists to finally find compelling evidence for ATP's role in muscle energetics. Lange (1955) subsequently reported a reduction in ATP concentration in the absence of a fall in PCr during contractions in frog muscle. Cain and Davies (1962) later reported a measurable change in ATP concentration during a single muscle contraction when creatine kinase was pharmacologically blocked using 1, fluoro-2, 4-dinitrobenzene. This is now held as definitive evidence that ATP is the primary energy source for muscular contractions. Hill was thus instrumental in driving the muscle physiology revolution forward,

resulting in a model of muscle bioenergetics that remains largely unchanged today. Moreover, the earlier concept advanced by Hill and others (of 'immediate' energy used in contraction, followed by recovery processes to restore it) was correct, requiring only revision of the specific metabolites involved. That concept finds its modern expression in the exercise physiology laboratory through the field of $\dot{V}_{O_2}$ kinetics. One of Hill's enduring strengths, we again suggest, was his ability to synthesise accurate theoretical concepts from often incomplete or limited experimental outcomes and specific details.

## The intensity dependence of the fatigue process

Hill noticed that the bioenergetics underpinning the speed–duration curve, within what we now recognise as the severe-intensity domain, would not suffice to adequately describe the highest intensity races:

'It is obvious, therefore, that we can not pursue our argument below times of about 50 s, that the maximum speed is limited by quite other factors than the amount of energy available.' (Hill, 1925c; p 415).

The inability of the hyperbolic model to characterise data below 2 min is obvious from Fig. 2*A*, which is the result of the inability to accumulate the distance represented by the curvature constant parameter, $D'$, above the critical speed before the race is completed. This is also why experimental predicting trials resulting in task failure in less than 2 min result in poor hyperbolic model fits (Bishop et al., 1998; Hill et al., 2002). The observations of Hill (1925c) regarding the limits to which $\dot{V}_{O_2}$ and the 'O$_2$ debt' could be used to explain performance therefore broadly agree with the scope of the severe-intensity domain we understand today (Jones et al., 2019).

Hill also recognised that the speed–duration relationship he presented up to ∼26 min did not hold for longer race durations, implying that the description of the speed–duration relationship, and its physiological interpretation, could not rely on a single model formulation:

'In fig. 1 we saw that the speed fell to what seemed to be practically a constant level towards the right of the diagram: this fall represents the initial factor in fatigue. On the logarithmic scale, however, where the longer times are compressed together, the curve continues to fall throughout its length. This later fall is due to factors quite different from those discussed above. Consideration merely of oxygen intake and oxygen debt will not suffice to explain the continued fall of the curve.' (Hill, 1925c; pp 421–422).

Hill concluded not only that exercise-induced fatigue is multifaceted, but also that the speed–duration curve could reveal these characteristics. These ideas are consistent with those of the contemporary exercise intensity domains

schema (Burnley & Jones, 2018; Whipp, 1994). However, the notion that physiological responses could be stratified in this way developed independently of models of athletic performance, first as a means of classifying physical work capabilities in the context of industry (Wells et al., 1957), and later as a means of understanding the physiology and pathophysiology of exercise in clinical populations (Wasserman et al., 1967). The discovery of the $\dot{V}_{O_2}$ slow component and its emergence at work rates above the lactate threshold (Whipp & Wasserman, 1972) set the stage for the marriage of the physiological response profile and the speed– or power–duration relationship. This was demonstrated first by Poole et al. (1988), who noted that the critical power asymptote separated steady state and non-steady state domains (heavy- and severe-intensity exercise, respectively). Subsequent work has shown that the metabolic and neuromuscular fatigue profiles are starkly different below the critical power/speed compared to above it (Black et al., 2017; Burnley et al., 2012; Jones, Wilkerson, DiMenna et al., 2008; Vanhatalo et al., 2016). Recent evidence has also suggested that fatigue processes and perceptual responses differ between moderate and heavy exercise (i.e. exercise performed below and above the lactate threshold) (Black et al., 2017; Brownstein et al., 2022; Iannetta et al., 2022). This supports the view first expressed by Hill (1925c) that distinct physiological processes are required to explain performance differences across the exercise intensity spectrum:

'The continued fall in the curve, as the effort is prolonged, is probably due either to the exhaustion of the material of the muscle, or to the incidental disturbances which may make a man stop before his muscular system has reached its limit.' (Hill, 1925c; p. 422).

Hill goes on to estimate that 'a man of average size' would utilise 300 g of glycogen in 1 h during a race (the 'material of the muscle'), implying that, within a few hours, the runner's glycogen store would be used up. Hill provides no references or workings for these figures, but it is reasonable to suggest that he derived them from the work of Zuntz (1901) and Krogh and Lindhard (1920), assuming that, at racing speeds, carbohydrate was the predominant fuel source for oxidative metabolism. The metabolic role of glycogen was by this point well established, following its discovery in 1858 (Philp et al., 2012), although direct measurements of glycogen utilisation in exercising human muscle would not be published until the 1960s (Bergstrom & Hultman, 1966). Nevertheless, the utilisation of 300 g of glycogen in 1 h implies that, at an RER of 0.9, the athlete would need to sustain a submaximal $\dot{V}_{O_2}$ of ∼3.9 L min$^{-1}$ (∼56 mL kg$^{-1}$ min$^{-1}$, assuming a body mass of 70 kg, an energy density of CHO of 16 kJ g$^{-1}$ and O$_2$ consumption providing 20.7 kJ L$^{-1}$). Elite athletes of the period were probably able to sustain such demands comfortably (e.g. Paavo Nurmi).

Hill (1925c) subsequently highlights the necessity of consuming carbohydrates 'as the effort continues' during prolonged races. Hill does not cite a reference to support that statement, but he made it a year after the paper of Levine and colleagues (1924) reported a positive correlation between the subjective condition of runners finishing the Boston marathon and their blood glucose concentrations. Moreover, those who had ingested carbohydrates before or during the race better maintained their blood glucose concentration at the finish line. The potential benefit of consuming carbohydrates to sustain endurance activity was already appreciated at the time, in part because of the well-publicised use of Kendal Mint Cake in the 1914–1917 Antarctic expedition led by Sir Ernest Shackleton. However, experimental studies of carbohydrate ingestion during exercise in humans did not take place until the 1970s (Ahlborg & Felig, 1976; Costill et al., 1973; Ivy et al., 1979), with the seminal study of Coyle et al. (1986) showing that exogenous carbohydrates replaced, rather than spared, muscle glycogen as a fuel source during prolonged exercise. Carbohydrate supplementation also played a central role in attempts to break the 2 h marathon barrier (Jones et al., 2021), with the scientific team involved demonstrating how performance capabilities could be maintained during heavy exercise lasting 2 h using now-standard methods of carbohydrate ingestion (Clark et al., 2019). Hill's view of the importance of carbohydrate feeding during exercise was decades ahead of its experimental confirmation.

Hill expressed reservations about the reliability of world record data for running beyond ~10 miles from fig. 6 in Hill (1925c). Remarkably, however, Hill's plot of the data up to 100 miles in 1925 is also consistent with the speed–duration profile during modern ultramarathon races: the world record lines are almost parallel across all durations (Fig. 4). Additionally, the profile reiterates that no single smooth function accurately characterises the speed–duration relationship across the continuum of endurance running competition. Figure 4 shows that there is a marked discontinuity in the curve between the marathon and races of 50 km and beyond, as shown by the arrow. That is, ultradistance races are disproportionately slower than distance races up to and including the marathon. The basis of this discontinuity is not clear, but record-breaking marathon runners rarely compete in ultramarathon events. Ultramarathon runners, in contrast, tend to specialise at those distances, competing in events where race time is often a secondary consideration given the course profile and difficulty (e.g. the Comrades Marathon, a race in KwaZulu-Natal over ~54 miles between Durban and Pietermaritzburg in South Africa, which alternates the starting point each year to provide an uphill or downhill challenge).

## The energetics of cycling

The fact that the speed–duration relationship followed the same qualitative pattern for swimming as for running (Fig. 1) provided further evidence to Hill that the curves had common physiological causes. Hill (1925c) acknowledged that this observation was not new in 1925: Kennelly (1906) had shown that log–log plots of the speed–duration relationship in a variety of modalities had approximately the same slopes. Even the performance of horses conformed to this same pattern. Hill (1925c) interpreted the fatiguability in cycling in light of the relatively small loss of speed as race duration was extended as follows:

'It is obvious at once that neither of these two curves [cycling and horse racing] falls anything like so rapidly as does that of a running man; fatigue does not so soon set in; the amount of energy expended at the highest speed must be much less than in a running man. This conclusion, indeed, is obvious to anyone who has tried to ride a bicycle fast. It is impossible to exhaust oneself rapidly on a bicycle' (Hill, 1925c; p. 419).

There is now overwhelming evidence to the contrary! The modest fall in speed as race duration increased led Hill to assume that there was a limited degree of fatigue development during cycling compared to running. However, it is the power output rather than the speed that determines the energetic demands of a cycling task. Prior to 1925, the fastest cycling speed at which $\dot{V}_{O_2}$ had been measured was 21 km h$^{-1}$ (Zuntz, 1899), less than half the slowest record speed plotted by Hill (1925c). Cycling's speed–$\dot{V}_{O_2}$ relationship was not comprehensively examined until Griffith Pugh used a saloon car as a mobile laboratory at Radlett Aerodrome in the 1970s

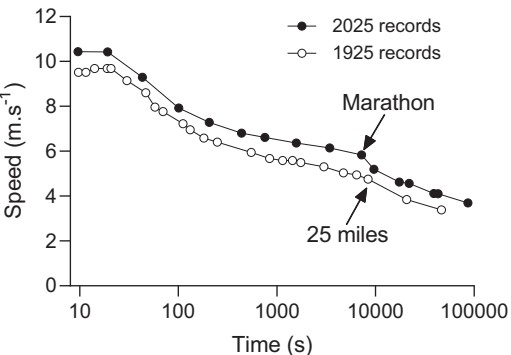

**Figure 4. Male running world records**
Male running world records up to and including 100 miles (1925) and up to 24 h (2025). Note that the profile of both data sets is almost identical, with a noticeable decrease in speed beyond the marathon/25 miles. Both sets of records include the modern standard marathon distance, introduced in the London Olympics so that the Royal Family could observe the finish at Windsor Castle, and standardised in Paris 1924 at 26 miles 385 yards (42.2 km).

(Pugh, 1974). At speeds above ~8 m s$^{-1}$ (29 km h$^{-1}$) air resistance became the dominant retarding factor, and the energetic demand increased as the cube of cycling speed.

Using the equation developed by di Prampero et al. (1979), we have calculated the approximate power requirements of the records used by Hill (1925c). These ranged from 422 W for the longest record (514.7 s, 12.6 m s$^{-1}$) to 974 W for the shortest (24.1 s, 16.9 m s$^{-1}$), assuming a standard racing position and the same rolling resistance used by di Prampero et al. (1979). In other words, the modest 25% decrease in speed Hill noted would have been associated with a 57% decrease in power output and, naturally, a similar decrease in energetic demand for the longer record performance (Fig. 5). Although Hill (1925c) did not make such a connection, 3 years later he published calculations on the air resistance of running based on wind tunnel experiments using a mannequin (Hill, 1928). Hill observed that the air resistance a runner experiences is small, but not negligible. He presented a table in which he calculated that, to equal the 10 mile world record, 40 W of muscular power would be needed just to overcome air resistance. Moreover, he estimated that such a requirement would be equivalent to running on a gradient of ~ 1%. Measurements made ~70 years later confirmed this (Jones & Doust, 1996). He also appreciated that because air resistance increases as a square of the windspeed it '… explains the difficulty of doing 'good' times on a windy day'. Because cycling is performed at more than double the speed of running, every day is a windy day, and the physiological requirements scale accordingly.

### Sex differences in human performance

The final exercise modality Hill (1925c) examined in detail was swimming, noting that:

'For a given time of swimming the maximum speed for a woman appears, throughout the curves, to be almost exactly 84 to 85 per cent. of that for a man … Women are well adapted to swimming: their skill in swimming is presumably just as great as that of men; the difference in the maximum speed for any given time can be a matter only of the amount of power available.' (Hill, 1925c; p. 415).

These swimming speed–duration curves provide the most robust sex comparison from Hill's data set, and using the hyperbolic model to derive the critical speed and $D'$ parameters shows that, as Hill suggested, the curves were the same shape ($D'$ ~29 m in each case), with the difference between them being the asymptote (1.21 m s$^{-1}$ for men, 1.02 m s$^{-1}$ for women – 84% of the value for men). For swimming, therefore, the differences in these record curves appear to be a result of factors exclusively associated with critical speed. Throughout the 20th century, the gap between male and female performance rapidly declined as the events that women were allowed to compete in increased and the participation of women in global sports events grew (Whipp & Ward, 1992b). However, that decline did not result in any crossover, and the performance gap remains at ~8–12% in athletics and ~5–13% in long course swimming (Sandbakk et al., 2018). The performance of both sexes is, unsurprisingly, substantially better than in 1925, but the sex gap in modern swimming records is significantly smaller for 800 and 1500 m than for shorter distances (Fig. 6). As a result, the calculated $D'$ from modern records is 29.4 m for men and 19.8 m for women, and the critical speeds are 1.69 m s$^{-1}$ and 1.61 m s$^{-1}$ for men and women, respectively (a difference of 5%). As mentioned above, because these curves are composed of times from several

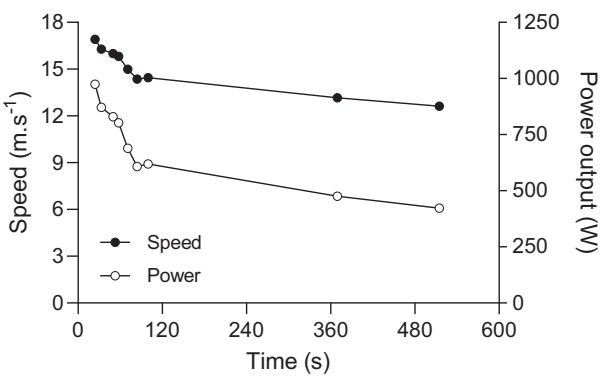

**Figure 5. Cycling speed and power profiles for unpaced records**
Cycling speed and calculated power profiles for unpaced records in 1925. Note the modest fall in speed but the substantial fall in calculated power output as trial duration increases.

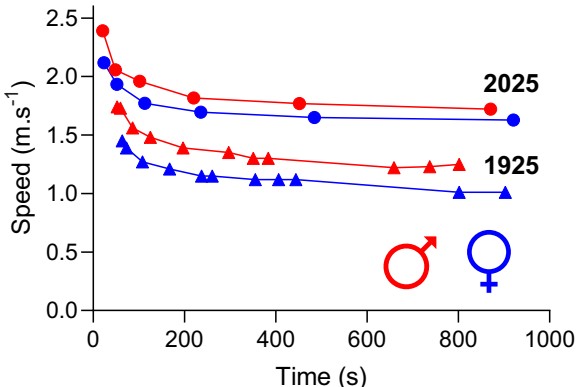

**Figure 6. Freestyle swimming world records for men and women**
Freestyle swimming world records for men (red symbols) and women (blue symbols) for both 1925 (triangles) and 2025 (circles). The profiles for all four sets are similar, but the 2025 records for women is substantially better than the men of 1925. Nevertheless, a sex difference in performance is evident for both record eras.

different athletes, we should be cautious about reading too much into them.

The world records for running in women presented by Hill (1925c) extend to no more than 200 s (1000 yards). The attitude of sporting authorities towards women competing in athletic events at the time was often hostile. The most notorious example of such hostility is found in the contemporaneous press accounts of the women's 800 m at the 1928 Amsterdam Olympics. Those reports suggested that half the field failed to finish, and several of those who did finish collapsed after crossing the line. This prompted the IAAF to ban women from racing distances of 800 m or longer, and these did not reappear at the Olympics until 1960. However, those press statements were fiction, including reporting the fate of 'eleven wretched women' when only nine competed, all of whom finished the race (Emery, 1985). Parity in participation between men and women was not achieved until the Olympics in Tokyo in 2021, and the scientific study of female performance still lags that of males (Elliott-Sale et al., 2021). Nonetheless, Hill's original interpretation that the speed–duration curves in men and women are qualitatively similar is supported by recent evidence demonstrating no sex differences in the power–duration relationship parameters normalised to the power output at $\dot{V}_{O_2max}$ (Ansdell et al., 2020).

### Pacing strategy

Having completed his description of world records, Hill (1925c) turned his attention to some other performance implications of his ideas, including optimal movement frequency and the considerations of internal and external work. Much of this discussion is based on the overestimation of the size of the $O_2$ deficit and oxygen requirement (predicting, for example, that running on the spot at 280 steps per minute required a $\dot{V}_{O_2}$ of 24 L min$^{-1}$; Hill [1925c], p. 425). However, using the same concepts, Hill suggested that the optimal performance strategy was to maintain an even pace for most race durations. But Hill then considered the potential physiological benefits of pacing relatively short races with a fast-start strategy:

'There may, indeed, be advantages in starting rather faster than the average speed which it is intended to maintain. The sooner the respiration and circulation are driven up to their maximum values, the greater will be the amount of oxygen taken in by the body, the greater the amount expendible [sic] during the race. It is a common practice in mile races to start very fast and to settle down later to the uniform speed: this may have a physiological basis in the quickening up of circulation and respiration achieved thereby.' (Hill, 1925c, p. 426).

This prediction was not systematically studied for more than 60 years (Foster et al., 1993), and subsequent studies demonstrated that short-duration performance (of ≤3 min) can indeed benefit from a fast start pacing strategy (Bishop et al., 2002; van Ingen Schenau et al., 1994). The mechanism underpinning this enhancement is not clear, but the $\dot{V}_{O_2}$ demand is initially high and then declines compared to an even paced effort. This fast start strategy initially causes the $\dot{V}_{O_2}$ response to rise more rapidly: a finding common to those studies showing improved performance (Bailey et al., 2011; Jones, Wilkerson, Vanhatalo et al., 2008; Sandals et al., 2006; Wood et al., 2014). Even when Hill's initial premise was incorrect, he was able to use first principles to produce accurate physiological predictions which were decades ahead of their experimental verification.

### Jumping performance analysis

The final section of Hill's papers dealt with more mechanical matters, following Hill's own observations from timing athletes in flight using a stopwatch. Using the equations of projectile motion, Hill was able to calculate the displacement of the centre of mass of athletes and suggest how to optimise jumping records. Hill calculated that the high jump record holder of 1925 had a centre of mass of ∼0.91 m standing still, and that to achieve the record he would need to raise it 1.1 m during the jump, requiring a 0.96 s flight time. This was plausible because the athletes Hill measured had cleared 1.5 m with flight times of 0.80 s. Hill further suggested that, to improve high jump performance, athletes should adopt a posture that allowed the body to clear the bar despite the centre of mass always being below it:

'Now, paradoxical as it may seem, it is possible for an object to pass over a bar while its centre of gravity passes beneath; every particle in the object may go over the bar and yet the whole time its centre of gravity may be below. A rope running over a pulley and falling the other side is an obvious example. It is conceivable that by suitable contortions the more accomplished high-jumpers may clear the bar without getting their centres of gravity above or appreciably above it.' (Hill, 1925c; p 427).

In the mid-1960s, high-jumpers Debbie Brill and Dick Fosbury independently pioneered the technique now used by high jumpers worldwide, the essential elements of which Hill had predicted 40 years earlier. The now famous 'Fosbury Flop' has also been termed the 'Brill Bend', which seems more appropriate given the trunk mechanics involved in performing the technique.

### Conclusions

Much has been written about the legacy of Hill's life and works in physiology, biophysics and the role of science in society (Katz, 1978; Bassett, 2002; Barclay & Curtin,

2022). We suggest that his work on the physiology of human performance is unique in that most of the topics he wrote about in that context remain active areas of research today. The speed– or power–duration relationships for the exercise modalities Hill presented are used extensively in athlete profiling (Leo et al., 2021; Pugh et al., 2022) and have also been applied to patient populations (Craig et al., 2019; Mezzani et al., 2010; Neder et al., 2000). That the character of the speed–duration relationship is common to all species so far studied demonstrates that its bioenergetic basis is hundreds of millions of years old (Burnley, 2023). The ubiquity of the relationship means that experiments can be conducted in animal models that would not be possible in exercising humans, substantially enhancing our mechanistic understanding of critical speed (Copp et al., 2010, 2013).

Hill's analysis of athletic records in the mid-1920s and their physiological bases stands up to modern scrutiny remarkably well. The analysis was performed in the middle of rapid progress in the understanding of muscle physiology, and Hill had only begun to systematically study exercise a few years before his lecture. Many of the physiological concepts we now take for granted (such as threshold-related phenomena, as well as the resulting non-linearities in the physiological response profiles) were largely unknown [but see Briggs (1920), rediscovered by Zoladz and Grassi (2020)]. Moreover, most of Hill's interpretations of human performance still hold, at least in outline; some (such as carbohydrate feeding during exercise, pacing strategy and jumping mechanics) were decades ahead of their time, whereas others, such as his examination of the female speed–duration relationship, and the similarities and differences between the sexes, remains a topic of intense investigation. In reaching their centenary, therefore, the surprising feature of Hill's papers is not how well they have aged, but how relevant they remain.

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

## Additional information

### Competing interests

The authors declare that they have no competing interests.

### Author contributions

All authors developed the review's concept. M.B. wrote the first draft and rewrote the manuscript following coauthor edits. A.V., D.C.P. and A.M.J. all edited each manuscript revision. All authors approved the final version of the manuscript submitted for publication.

### Funding

This review was not externally funded.

### Keywords

endurance performance, exercise, oxidative metabolism

## Supporting information

Additional supporting information can be found online in the Supporting Information section at the end of the HTML view of the article. Supporting information files available:

**Peer Review History**

