## [Peer Review History · The Journal of Physiology]

Blue Plaque Review Series: A.V. Hill, athletic records, and the birth of exercise physiology

Mark Burnley, Anni Vanhatalo, David C Poole, and Andrew M Jones
DOI: 10.1113/JP288130

Corresponding author(s): Mark Burnley (M.Burnley@lboro.ac.uk)

The following individual(s) involved in review of this submission have agreed to reveal their identity: L. Bruce Gladden (Referee #1); Pietro Enrico di Prampero (Referee #3)

Review Timeline:

Submission Date:	05-Dec-2024
Editorial Decision:	14-Jan-2025
Revision Received:	21-Jan-2025
Editorial Decision:	03-Feb-2025
Revision Received:	04-Feb-2025
Accepted:	06-Feb-2025

Senior Editor: Laura Bennet

Reviewing Editor: Bruno Grassi

Transaction Report:

Dear Dr Burnley,

Re: JP-TR-2024-288130 "Blue Plaque Review Series: A.V. Hill, athletic records, and the birth of exercise physiology" by Mark Burnley, Anni Vanhatalo, David C Poole, and Andrew M Jones

Thank you for submitting your manuscript to The Journal of Physiology. It has been assessed by a Reviewing Editor and by 3 expert referees and we are pleased to tell you that it is acceptable for publication following satisfactory revision.

ABSTRACT FIGURES: Authors may use The Journal's premium BioRender account to create/redraw their Abstract Figures (and any other suitable schematic figure). Information on how to access this account is here: <https://physoc.onlinelibrary.wiley.com/journal/14697793/biorender-access>.

REVISION CHECKLIST: Upload a full Response to Referees file. To create your 'Response to Referees' copy all the reports, including any comments from the Senior and Reviewing Editors, into a Microsoft Word, or similar, file and respond to each point, using font or background colour to distinguish comments and responses and upload as the required file type.

We look forward to receiving your revised submission.

Yours sincerely,

Laura Bennet
Senior Editor

EDITOR COMMENTS

Reviewing Editor:

Comments to the Author (Required):

This is an historical review based on some papers published by Nobel Laureate AV Hill 100 years ago, mainly relating to an analysis of world records during athletic performance, with inferences on physiological mechanisms. The authors do an excellent job in convincing the reader that AV Hill's inferences on several aspects of exercise physiology are still valid today, although they were formulated years or even decades before experimental evidence eventually confirmed them. The manuscript was evaluated by three very expert reviewers, who expressed enthusiasm in the manuscript, but raised several specific and relatively minor suggestions for changes and/or integration, which should be duly taken in consideration by the authors.

Please also see 'Required Items' below.

As per Editorial Office correspondence with author David Poole, we recommend you change the title of the article (in the revised Word file) to: 'Blue Plaque Review Series: A.V. Hill, athletic records, and the birth of exercise physiology'

REFEREE COMMENTS

Referee #1:

This is another among several interesting explorations of AV Hill's work that underpins much of basic exercise physiology. Overall, a nice read.

Here are some suggestions for consideration.

Perhaps somewhere among lines 69-91, it would be of interest to note something about nonlinear curve fitting and computers. Most, if not all, curve fitting programs that are readily available today (e.g., SigmaPlot, GraphPad Prism, OriginLab, KaleidaGraph) use the Levenberg-Marquardt algorithm to quickly provide a plethora of nonlinear plots if the user wishes. Apparently, Levenberg first published the algorithm in 1944 and then it was rediscovered or amplified by Marquardt in 1963. Many of today's young researchers are likely blissfully unaware of what it was like not to have this capability at their fingertips since many of these computer packages only became widespread in the 1990s.

L126-133 - perhaps somewhere note that Hill's fit was likely a visually determined nonlinear least squares fit? Maybe visually interpolated?

L205 - Perhaps reword "borders on the absurd".

L208-209 - Both Lohman and, again, Fiske and SubbaRow deduced the structure of ATP independently in 1929.

L224-227 - The establishment of ATP as the direct source of energy for muscle contraction is credited to Cain and Davies (1962) who poisoned creatine kinase with 2,4-DNFB. However, usually overlooked is that Lange (1955) showed decreases in [ATP] and increases in [ADP] due to muscle contraction with a combination of IAA and nitrogen mustard. - So, perhaps credit to both?

L235 - Perhaps "often incomplete or limited experimental details."

L251 - Unclear wording - "why experimental predicting trials shorter than 2 min result in..."

L297-298 - As late as 1923, Hill apparently thought that carbohydrate was the sole fuel for exercise metabolism. Was that still his thinking in 1925?

L397 - "curves provide"

L844 - Please reword "whose trajectory which results in". Slow component is not a "whose" and "which results" is confusing.

Referee #2:

General Comments

This is a very well written, interesting and informative brief review A. V. Hill's work exploring the physiology of exercise performance, on the 100th anniversary of some of the author's most well-known works. The interest in this review should not be limited to exercise physiologists, but to anyone interested in the historical development of contemporary research in exercise sciences. I congratulate the authors on an interesting and insightful manuscript. I have only a few general thoughts and specific comments for the authors to consider.

1. In Fig 2 the hyperbolic model deviates significantly from the data in durations below ~3 minutes. I may be mistaken, but visual inspection suggests that the curve is not a best fit to the data. The total summed error is not evenly distributed above and below the fitted curve. Please can the authors check that this is truly the best hyperbolic fit.
2. I recommend that the authors be very specific about definition of some terms, that have perhaps changed over time. For example, line 104 introduces the term "maximal aerobic power" and line 116 introduces "aerobic speed". Similarly line 116 uses the term "maximal speed" and line 122 introduces "maximal instantaneous sprinting speed". Some of these terms may relate to similar concepts, but some do not. I'd urge the authors to review the entire manuscript and be very specific with defining new terms when they are introduced. It may be preferred to include a glossary, where terms, and important differences between similar sounding terms, can be described in detail without losing the flow of the main text.
3. Lines 151-155. I would challenge the authors' assertion here that "The contemporary understanding is... almost identical to this interpretation". Hill's description in the preceding quote is based on available energetics e.g., VO₂max and a theoretic maximal size of the oxygen deficit (using current terminology). I believe that our current understanding of exercise durations over those considered here (~1-15 min) is better described by a term related to VO₂max (as in Hill's statement) and an accumulation of factors that limit muscle contraction, such as intramuscular variables that induce muscle fatigue, Pi, H⁺ and/or cause neural inhibition of muscle activation. I think the distinction between Hill's "account withdrawal" model (a limitation of available energy stores) and the more recent data supporting fatigue induction (a limitation in the ability to use available energy sources) is a significant divergence between then and now and is essential to our current understanding of exercise limitation. The latter is supported by studies that show that substrate level energy stores remain at intolerance of large muscle mass exercise, such as Cannon et al. (doi:10.1152/jappphysiol.00510.2013.), or models showing that the characteristics of the power-duration relationship can be predicted by intramuscular fatigue-related metabolites (doi:10.1152/jappphysiol.00745.2020). I recommend that this distinction be explored in more detail here. This issue is also revisited on lines 177-180, where a more specific phrasing about "fatigue induction" model could be used. This view is closer to the Hill quote on line 291.
4. Lines 426-429. Given the commonly posed argument that women are "less fatigable" than men, could a few sentences be added here about the physiological mechanisms underlying the (absence of) differences in normalized power-duration curves between men and women? I'm not suggesting a detailed description, but a broad muscle energetic context, of the sort provided earlier in the manuscript, to once again frame the male/female performance records in a physiological basis.

5. Lines 449-453. For me, this section is misleading. Reaching a greater VO₂ more quickly with a fast-start strategy does not necessarily equate to "faster VO₂ kinetics at exercise onset". In the studies of Sandals et al., 2006, Jones et al., 2008, Bailey et al. 2011 and Wood et al 2014, the faster start causes a greater percentage of VO₂max to be attained, and/or attained more rapidly, such that a mean response time is reduced. But this experimental method does not account for the energy demand, which is also greater at the beginning of the fast start strategy. Therefore, without knowledge of the energy demand, then it is not possible to infer that the time constant of VO₂ is actually lower ("faster VO₂ kinetics at exercise onset") or whether the "account withdrawal" is, in fact, lower (which is the assumption of faster VO₂ kinetics). The fast start may confer a performance advantage, but these studies do not provide evidence that this advantage is due to "faster VO₂ kinetics".

Specific Comments (line)

20 ... A. V. Hill authored three...

29 Importantly or impactfully? My feeling is that "importance" is up to the reader to decide, whereas impact is evident in the historical record.

30 ...characteristics that reflect...

32 Fatigue is such a laden term. Without the space in the abstract to develop it, perhaps the following would be a safer statement "...nature of mechanisms limiting exercise tolerance."

44 Perhaps "...British Association for the Advancement of Science..."

50 Perhaps "...of the 1922 Nobel Laureate..."

82 Can you infer, and state what it was, that Hill thought was "extremely interesting and suggestive" about Kennelly's paper?

128 Suggest "...the model error associated with events <3 min in..."

140 "...in which similar baseline physiological status exists..."

180-185 I can help but think that a reference to a Whipp (et al?) paper is warranted here, but I can't find the paper I am looking for, which shows the O₂ deficit during exercise that asymptotes above VO₂max. Whipp showed that the calculated accumulated O₂ deficit is constant for exercise that VO₂ asymptotes above VO₂max, and can be calculated from VO₂*tau (equation 3.20 in Ch 3. Of Jones and Poole VO₂ kinetics book).

195-200 Perhaps include citation here to doi:10.1007/s00421-003-0870-y

248 ...Figure 2A...

252-254 I'm not certain what is being referred to in this sentence. Is it only the <2 min data points or are you also referring to durations at the other end of the speed-duration plot? Up to 15 min or beyond 15 min? Please clarify.

261. Does Fig 1 in this quote related to Fig 1 in the current manuscript? I found the text is this quote rather abstract. Could some additional explanation be added to describe how this relates to Fig 1 (or 2) in the manuscript?

279 Re. "steady state and non-steady state domains". I was struck by the Hill quote on line 161 that used the thermodynamically correct term "dynamic equilibrium". You have an opportunity here to correct the literature on the use of the term "steady-state" in relation to these exercise intensity domains. "Steady-state" describes a condition where all system variables remain constant over time. It requires that at all variables in a system (the body, in this case) remain constant as time changes. There must be no change in mass or energy over time. This is not true for exercise mildly below CS/CP, where, for example, muscle glycogen is continually falling as exercise continues. Dynamic equilibrium better describes the condition of moderate and heavy intensity exercise, where typically measured variables are stable but this stability is maintained by changes internal to the system (body) that are not typically observed, such as mobilization of fatty acid and/or glucose from one tissue compartment for use in another.

286 You might consider including a description of how extreme duration events fit into this scheme e.g., doi:10.1126/sciadv.aaw0341.

300 ... glycogen utilisation in exercising human muscle would...

302 Why is it reasonable to use an RER of 0.9? This seems low for "racing speeds", even for longer duration races such as the marathon.

327 ... from Figure 6 in Hill (1925c)."

334 Suggest to end the sentence at "...compete in ultramarathon events. Ultramarathon runners...". There are likely many reasons for this and there is no advantage to speculation here.

397 ...curves provide the...

897 Suggest to include a legend within Figure 6 to explain the different symbols and colours. Is color necessary?

Referee #3:

This is a very interesting review of A.V. Hill's 100 years old analysis of world records in some forms of human locomotion, as reported in his three fascinating papers published in 1925. The authors deal mainly with running and, to a lesser extent, with swimming and cycling and clearly show that Hill's outstanding approach to the energetics of human locomotion is essentially supported by 100 years of detailed experimental studies in this and related matters. The authors are to be praised for their profound analysis of Hill's ideas, as well as for their detailed review of 100 years literature in these matters.

A few comments are reported below for the authors' convenience.

General.

The legend to figure 3, lines 847 - 848, reports: "In both plots it is assumed that the primary VO₂ gain is 10 mL.min⁻¹. W⁻¹.....". In fact, this is the efficiency of the transformation of metabolic to mechanical power. Indeed, assuming a Respiratory Quotient of 0.95, the consumption of 1 L O₂ in the human body yields 20.9 kJ; as such a "primary gain" of 10 mL.min⁻¹.W⁻¹ is equivalent to a mechanical efficiency of 0.287. I would therefore suggest the term "mechanical efficiency" rather than "primary gain". Furthermore the efficiency, as estimated from the "steady state" values (for t ≈ 840 s) of net VO₂ (above resting) and CP of panels B and D of figure 3, assuming an RQ of 0.95, is ≈ 0.21, i.e. essentially smaller than 0.287.

Again referring to Fig. 3, panels B and D are not coherent with the effects of the acceleration occurring at the beginning of the shortest sprint distances (100 to 400 m). In this case the metabolic power increases abruptly in the first few seconds of the run, and declines in its second part, whereas the mechanical power, after the initial abrupt rise, remains substantially constant, thus leading to a substantial increase of the mechanical efficiency in the second part of the run. This state of affairs is described in detail in some recent studies (two of which are reported below) that have escaped the authors' attention. I do not suggest that the authors go into such details, particularly so because they apply only to the very short distances mentioned above. They may, if they wish, add a very short note on this problem that, in any case does not deserve any detailed discussion.

- The energy cost of sprint running and the energy balance of current world records from 100 to 5000 m. P.E. di Prampero, C. Osgnach. In: Biomechanics of Training and Testing. J.-B. Morin, P. Samozino eds. (Springer International), Ch. 12, PP 269 - 297, 2018.

- Mechanical and metabolic power in accelerated running - Part I: the 100-m dash. P.E. di Prampero, C. Osgnach, J.-B. Morin, P. Zamparo, G. Pavei. Eur. J. Appl. Physiol. DOI: <https://doi.org/10.1007/s00421-023-05236-x>.

Specific.

- P.7, line173: Jones et al. 2008: please specify whether a, or b;

- P. 11, line 282: Jones et al. 2008: please specify whether a, or b;

- P.17, line 453: : Jones et al. 2008: please specify whether a, or b;

- P.7, line175, Whipp and Ward, 1992: please specify whether a, or b;

- P.11, line405, Whipp and Ward, 1992: please specify whether a, or b;

- P.25, lines 665 and 669: please add a, or b;

- P.25, line 678: the reference to Katz and Katz (1999) is not mentioned in the text;

- P.26, line 704: the reference to Mitchell et al. (2018) is not mentioned in the text;
- P.29, lines 793 and 796: please add a, or b.

REQUIRED ITEMS

- Please include an Abstract Figure file, as well as the Figure Legend text within the main article file. The Abstract Figure is a piece of artwork designed to give readers an immediate understanding of the Review Article and should summarise the main conclusions. If possible, the image should be easily 'readable' from left to right or top to bottom. It should show the physiological relevance of the Review so readers can assess the importance and content of the article. Abstract Figures should not merely recapitulate other figures in the Review. Please try to keep the diagram as simple as possible and without superfluous information that may distract from the main conclusion of the Review. Abstract Figures must be provided by authors no later than the revised manuscript stage and should be uploaded as a separate file during online submission labelled as File Type 'Abstract Figure'. Please ensure that you include the figure legend in the main article file. All Abstract Figures will be sent to a professional illustrator for redrawing and you may be asked to approve the redrawn figure before your paper is accepted.

- Your MS must include a complete "Additional information section" with the following 4 headings and content:

Competing Interests: A statement regarding competing interests. If there are no competing interests, a statement to this effect must be included. All authors should disclose any conflict of interest in accordance with journal policy.

Author contributions: Each author should take responsibility for a particular section of the study and have contributed to writing the paper. Acquisition of funding, administrative support or the collection of data alone does not justify authorship; these contributions to the study should be listed in the Acknowledgements. Additional information such as 'X and Y have contributed equally to this work' may be added as a footnote on the title page.

It must be stated that all authors approved the final version of the manuscript and that all persons designated as authors qualify for authorship, and all those who qualify for authorship are listed.

Funding: Authors must indicate all sources of funding, including grant numbers. If authors have not received funding, this must be stated.

It is the responsibility of authors funded by RCUK to adhere to their policy regarding funding sources and underlying research material. The policy requires funding information to be included within the acknowledgement section of a paper. Guidance on how to acknowledge funding information is provided by the Research Information Network. The policy also requires all research papers, if applicable, to include a statement on how any underlying research materials, such as data, samples or models, can be accessed. However, the policy does not require that the data must be made open. If there are considered to be good or compelling reasons to protect access to the data, for example commercial confidentiality or legitimate sensitivities around data derived from potentially identifiable human participants, these should be included in the statement.

Acknowledgements: Acknowledgements should be the minimum consistent with courtesy. The wording of acknowledgements of scientific assistance or advice must have been seen and approved by the persons concerned. This section should not include details of funding.

- Please upload separate high quality figure files via the submission form.

- Author profile(s) must be uploaded via the submission form. Authors should submit a short biography (no more than 100 words for one author or 150 words in total for two authors) and a portrait photograph of the two leading authors on the paper. These should be uploaded and clearly labelled together in a Word document with the revised version of the manuscript. Any standard image format for the photograph is acceptable, but the resolution should be at least 300 DPI and preferably more. A group photograph of all authors is also acceptable, providing the biography for the whole group does not exceed 150 words.

- It is the authors' responsibility to obtain any necessary permissions to reproduce previously published material and to list these within the main article file. For information, please see: https://jp.msubmit.net/cgi-bin/main.plex?form_type=display_requirements#permissions.

END OF COMMENTS

This is a very interesting review of A.V. Hill's 100 years old analysis of world records in some forms of human locomotion, as reported in his three fascinating papers published in 1925. The authors deal mainly with running and, to a lesser extent with swimming and cycling and clearly show that Hill's outstanding approach to the energetics of human locomotion is essentially supported by 100 years of detailed experimental studies in this and related matters. The authors are to be praised for their profound analysis of Hill's ideas, as well as for their detailed review of 100 years literature in these matters. A few comments are reported below for the authors' convenience.

General.

The legend to figure 3, lines 847 – 848, reports: "In both plots it is assumed that the primary VO_2 gain is $10 \text{ mL}\cdot\text{min}^{-1}\cdot\text{W}^{-1}\dots\dots$ ". In fact, this is the efficiency of the transformation of metabolic to mechanical power. Indeed, assuming a Respiratory Quotient of 0.95, the consumption of 1 L O_2 in the human body yields 20.9 kJ; as such a "primary gain" of $10 \text{ mL}\cdot\text{min}^{-1}\cdot\text{W}^{-1}$ is equivalent to a mechanical efficiency of 0.287. I would therefore suggest the term "mechanical efficiency" rather than "primary gain". Furthermore the efficiency, as estimated from the "steady state" values (for $t \approx 840 \text{ s}$) of net VO_2 (above resting) and CP of panels B and D of figure 3, assuming an RQ of 0.95, is ≈ 0.21 , i.e. essentially smaller than 0.287.

Again referring to Fig. 3, panels B and D are not coherent with the effects of the acceleration occurring at the beginning of the shortest sprint distances (100 to 400 m). In this case the metabolic power increases abruptly in the first few seconds of the run, and declines in its second part, whereas the mechanical power, after the initial abrupt rise, remains substantially constant, thus leading to a substantial increase of the mechanical efficiency in the second part of the run. This state of affairs is described in detail in some recent studies (two of which are reported below) that have escaped the authors' attention. I do not suggest that the authors go into such details, particularly so because they apply only to the very short distances mentioned above. They may, if they wish, add a very short note on this problem that, in any case does not deserve any detailed discussion.

- The energy cost of sprint running and the energy balance of current world records from 100 to 5000 m. P.E. di Prampero, C. Osgnach. In: Biomechanics of Training and Testing. J.-B. Morin, P. Samozino eds. (Springer International), Ch. 12, PP 269 – 297, 2018.

- Mechanical and metabolic power in accelerated running – Part I: the 100-m dash. P.E. di Prampero, C. Osgnach, J.-B. Morin, P. Zamparo, G. Pavei. Eur. J. Appl. Physiol. DOI: <https://doi.org/10.1007/s00421-023-05236-x>.

Specific.

- P.7, line173: Jones et al. 2008: please specify whether a, or b;
- P. 11, line 282: Jones et al. 2008: please specify whether a, or b;
- P.17, line 453: : Jones et al. 2008: please specify whether a, or b;
- P.7, line175, Whipp and Ward, 1992: please specify whether a, or b;
- P.11, line405, Whipp and Ward, 1992: please specify whether a, or b;
- P.25, lines 665 and 669: please add a, or b;
- P.25, line 678: the reference to Katz and Katz (1999) is not mentioned in the text;
- P.26, line 704: the reference to Mitchell et al. (2018) is not mentioned in the text;
- P.29, lines 793 and 796: please add a, or b.

EDITOR COMMENTS

We thank the editor and reviewers for their positive comments on our manuscript and most helpful suggestions for improvement. Our responses are given in red font throughout.

Reviewing Editor:

Comments to the Author (Required):

This is an historical review based on some papers published by Nobel Laureate AV Hill 100 years ago, mainly relating to an analysis of world records during athletic performance, with inferences on physiological mechanisms. The authors do an excellent job in convincing the reader that AV Hill's inferences on several aspects of exercise physiology are still valid today, although they were formulated years or even decades before experimental evidence eventually confirmed them. The manuscript was evaluated by three very expert reviewers, who expressed enthusiasm in the manuscript, but raised several specific and relatively minor suggestions for changes and/or integration, which should be duly taken in consideration by the authors.

Please also see 'Required Items' below.

As per Editorial Office correspondence with author David Poole, we recommend you change the title of the article (in the revised Word file) to: 'Blue Plaque Review Series: A.V. Hill, athletic records, and the birth of exercise physiology'

Thank you, this has been changed in the revised manuscript.

REFEREE COMMENTS

Referee #1:

This is another among several interesting explorations of AV Hill's work that underpins much of basic exercise physiology. Overall, a nice read.

We thank you for these positive comments. We have revised the manuscript in line with your suggestions and believe that the clarity and potential impact of the revised version is much improved as a result.

Here are some suggestions for consideration.

Perhaps somewhere among lines 69-91, it would be of interest to note something about nonlinear curve fitting and computers. Most, if not all, curve fitting programs that are readily available today (e.g., SigmaPlot, GraphPad Prism, OriginLab, KaleidaGraph) use the Levenberg-Marquardt algorithm to quickly provide a plethora of nonlinear plots if the user wishes. Apparently, Levenberg first published the algorithm in 1944 and then it was rediscovered or amplified by Marquardt in 1963. Many of today's young researchers are

likely blissfully unaware of what it was like not to have this capability at their fingertips since many of these computer packages only became widespread in the 1990s.

Thank you, an excellent suggestion. We have added a short section to address this point:

“Least squares non-linear regression algorithms commonly used today were not developed until the mid-1940s (Levenberg, 1944; Marquardt, 1963). Furthermore, the Levenberg-Marquardt algorithm did not become widely available in statistical and graphing software for personal computers until the 1990s. In lieu of equations and the algorithm to fit them, Hill described the plots in Figure 1 thus:”

L126-133 - perhaps somewhere note that Hill's fit was likely a visually determined nonlinear least squares fit? Maybe visually interpolated?

This point might be best expressed from line 69. It is difficult to say what approach Hill took in drawing the original curves, because the precise method is not stated. However, we can say that it was a visual attempt to minimise the errors in the fitted curve, either by eye or with the aid of a French Curve. The truth is we don't know! We have added a clause to an earlier sentence. It now reads:

“Notably, the paper does not provide equations associated with any of the curves describing the speed-duration relationships, which would have been hand drawn, with the error about the fitted line determined by eye.”

We think going into greater detail than the final clause would be too speculative.

L205 - Perhaps reword "borders on the absurd".

Thank you, we have replaced this with “astonishing”

L208-209 - Both Lohman and, again, Fiske and Subbarow deduced the structure of ATP independently in 1929.

Thank you, we have added the following sentence and reference following mention of Lohmann's work:

“Fiske and Subbarow (1929) independently confirmed its existence soon afterwards.”
With “its” referring to ATP.

L224-227 - The establishment of ATP as the direct source of energy for muscle contraction is credited to Cain and Davies (1962) who poisoned creatine kinase with 2,4-DNFB. However, usually overlooked is that Lange (1955) showed decreases in [ATP] and increases in [ADP] due to muscle contraction with a combination of IAA and nitrogen mustard. - So, perhaps credit to both?

Thank you for highlighting this. We have added the Lange (1955) reference to the text. The full paragraph now reads:

“By the mid-1930s, several lines of evidence placed ATP as the likely primary energy donor during muscle contraction (for a comprehensive historical review, see Rall [2023]). Hill

(1950) challenged biochemists to finally find compelling evidence for ATP's role in muscle energetics. Lange (1955) subsequently reported a reduction in ATP concentration in the absence of a fall in PCr during contractions in frog muscle. Cain and Davies (1962) later reported a measurable change in ATP concentration during a single muscle contraction when creatine kinase was pharmacologically blocked using 1,fluoro-2,4-dinitrobenzene. This is now held as definitive evidence that ATP is the primary energy source for muscular contractions. Hill was thus instrumental in driving the muscle physiology revolution forward, resulting in a model of muscle bioenergetics which remains largely unchanged today. Moreover, the earlier concept advanced by Hill and others (of "immediate" energy used in contraction, followed by recovery processes to restore it) was correct, requiring only revision of the specific metabolites involved. That concept finds its modern expression in the exercise physiology laboratory through the field of $\dot{V}O_2$ kinetics. One of Hill's enduring strengths, we again suggest, was his ability to synthesise accurate theoretical concepts from often incomplete or limited experimental details."

Note that we have given Lange primacy for the discovery. Re-reading Cain and Davies, the omission of a citation to Lange's work is very surprising.

L235 - Perhaps "often incomplete or limited experimental details."

Thank you, changed.

L251 - Unclear wording - "why experimental predicting trials shorter than 2 min result in..."

Thank you, we have changed this to "why experimental predicting trials resulting in task failure in less than 2 min..."

L297-298 - As late as 1923, Hill apparently thought that carbohydrate was the sole fuel for exercise metabolism. Was that still his thinking in 1925?

This is difficult to say for certain, but our reading of his 1925 papers does not rule out fat as a source of energy during exercise: "After a very few hours, therefore, the whole glycogen supply of his body will be exhausted. The body, however, **does not readily use fat alone as a source of energy** (emphasis ours): disturbances may arise in the metabolism; it will be necessary to feed a man with carbohydrate as the effort continues." This suggests to us that Hill at least considered fat as part of the energy mix, but not without consequences: Specifically, what we now know to be ketoacidosis and a lower rate of energy transfer.

L397 - "curves provide"

Thank you, changed.

L844 - Please reword "whose trajectory which results in". Slow component is not a "whose" and "which results" is confusing.

Thank you, we have removed "whose trajectory". This results in a sentence that reads: "In panel B the $\dot{V}O_2$ responses include a slow component which results in the attainment of $\dot{V}O_{2max}$ and exercise intolerance..."

Referee #2:

General Comments

This is a very well written, interesting and informative brief review A. V. Hill's work exploring the physiology of exercise performance, on the 100th anniversary of some of the author's most well-known works. The interest in this review should not be limited to exercise physiologists, but to anyone interested in the historical development of contemporary research in exercise sciences. I congratulate the authors on an interesting and insightful manuscript. I have only a few general thoughts and specific comments for the authors to consider.

We would like to thank the reviewer for his/her positive review of our manuscript. We have incorporated the reviewer's insightful suggestions where possible and this has improved the manuscript substantially.

1.

In Fig 2 the hyperbolic model deviates significantly from the data in durations below ~3 minutes. I may be mistaken, but visual inspection suggests that the curve is not a best fit to the data. The total summed error is not evenly distributed above and below the fitted curve. Please can the authors check that this is truly the best hyperbolic fit.

The visual inspection is correct, but the explanation for the apparent deviation of the fitted line from the data is discussed within the manuscript. Simply stated, this deviation is a necessary consequence of the hyperbola being constrained to reach an infinite speed at time zero, and for any time greater than zero and any speed above CS at least 246 m of distance needs to be accumulated above CS (5.3 m/s) over that time period. In reality, this accumulation at normal running speeds requires at least 2 min, and for events shorter than this the curve will necessarily overestimate performance. This is why we do not fit the hyperbolic function to experimental trials in which task failure occurs in less than 2 min. In producing this figure, we also compared the 2-parameter critical speed curve to a 3-parameter curve, which includes a constant, k , allowing the time asymptote to freely vary. This model fits the data throughout, but at the cost of an extra fitting parameter. Analysis using Akaike's Information Criterion showed that the 2-parameter model was still the superior choice despite the correctly identified deviation. The plot you see in the paper is a result of fitting the hyperbolic function beyond its applicable range.

2. I recommend that the authors be very specific about definition of some terms, that have perhaps changed over time. For example, line 104 introduces the term "maximal aerobic power" and line 116 introduces "aerobic speed". Similarly line 116 uses the term "maximal speed" and line 122 introduces "maximal instantaneous sprinting speed". Some of these terms may relate to similar concepts, but some do not. I'd urge the authors to review the entire manuscript and be very specific with defining new terms when they are introduced. It may be preferred to include a glossary, where terms, and important differences between similar sounding terms, can be described in detail without losing the flow of the main text.

Thank you, and we fully agree about the careful use of terms. In our defence, the terms quoted above are not terms used by us (or, indeed, used subsequently), because they are terms specific to the mathematical models in question. We have clarified the use of "maximal

aerobic power” from Wilkie’s formulation, since this model used cycling rather than running. Everywhere else in the manuscript, power means cycling power and speed means running, swimming or rowing speed. We have checked the manuscript, and we don’t think there are any further issues to address.

3. Lines 151-155. I would challenge the authors' assertion here that "The contemporary understanding is... almost identical to this interpretation". Hill's description in the preceding quote is based on available energetics e.g., VO₂max and a theoretic maximal size of the oxygen deficit (using current terminology). I believe that our current understanding of exercise durations over those considered here (~1-15 min) is better described by a term related to VO₂max (as in Hill's statement) and an accumulation of factors that limit muscle contraction, such as intramuscular variables that induce muscle fatigue, Pi, H⁺ and/or cause neural inhibition of muscle activation. I think the distinction between Hill's "account withdrawal" model (a limitation of available energy stores) and the more recent data supporting fatigue induction (a limitation in the ability to use available energy sources) is a significant divergence between then and now and is essential to our current understanding of exercise limitation. The latter is supported by studies that show that substrate level energy stores remain at intolerance of large muscle mass exercise, such as Cannon et al. (doi:10.1152/jappphysiol.00510.2013.), or models showing that the characteristics of the power-duration relationship can be predicted by intramuscular fatigue-related metabolites (doi:10.1152/jappphysiol.00745.2020). I recommend that this distinction be explored in more detail here. This issue is also revisited on lines 177-180, where a more specific phrasing about "fatigue induction" model could be used. This view is closer to the Hill quote on line 291.

Thank you, we have revised the passages according to the reviewer’s suggestions, it now reads:

“The contemporary understanding of high-intensity exercise tolerance is similar to this interpretation, insofar as exercise tolerance during severe-intensity exercise (i.e., above the critical speed, the domain in which many Olympic athletic events take place) is determined, in part, by the interaction of maximal oxygen uptake ($\dot{V}O_{2\max}$) and the capacity for substrate-level phosphorylation (Burnley and Jones, 2007). However, modern theory deviates somewhat from Hill’s conceptual understanding in two key respects. First, the tolerable duration of exercise was thought to be the consequence of the accumulation of a maximal O₂ deficit, whereas it is now thought to be a function of both the depletion of high-energy phosphates and the accumulation of fatiguing metabolites (particularly inorganic phosphate and H⁺; Jones et al., 2008; Cannon et al., 2013; Korzeniewski & Rossiter, 2021). These changes, in turn, have neurophysiological and perceptual effects which may also limit exercise tolerance (Hureau et al., 2018). Second, in earlier works Hill interpreted the transition from steady state to non-steady state as occurring at the running speed associated with $\dot{V}O_{2\max}$ (Hill and Lupton, 1923; Figure 3), even using the phrase “critical speed” to describe it:

“Considering the case of running, there is clearly some critical speed for each individual, below which there is a genuine dynamic equilibrium, break-down being balanced by restoration, above which, however, the maximum oxygen intake is inadequate, lactic acid accumulating, a continuously increasing oxygen debt being incurred, fatigue and exhaustion setting in.” (Hill and Lupton, 1923; p. 151.)

In contrast, the modern understanding of these responses is that non-steady state behaviour emerges at a critical $\dot{V}O_2$ value (and critical running speed) substantially below $\dot{V}O_{2max}$ (Jones et al., 2010; Poole et al., 2016; Goulding and Marwood, 2023). At such speeds, in the severe intensity domain, $\dot{V}O_2$ increases progressively to attain $\dot{V}O_{2max}$ due to the development of the slow component of $\dot{V}O_2$ kinetics (Whipp and Mahler, 1980; Poole et al. 1988; Whipp, 1994). At the same time, muscle phosphocreatine (PCr) progressively falls, and inorganic phosphate progressively rises, indicating an obligatory energetic contribution from substrate-level phosphorylation (Jones et al., 2008; Vanhatalo et al., 2010; 2016). This loss of what Hill termed “dynamic equilibrium” at speeds below $\dot{V}O_{2max}$ but above the critical speed thus initiates a chain of events that ultimately limits exercise tolerance. It is important to note that “dynamic equilibrium” is a more thermodynamically precise description than “steady state”, because metabolically relevant variables, notably muscle glycogen (Black et al., 2016), continue to change under conditions of a steady state $\dot{V}O_2$ (below the critical speed).

Above the critical speed, the trajectory of the $\dot{V}O_2$ slow component determines when $\dot{V}O_{2max}$ is reached, with task failure occurring soon thereafter (Whipp and Ward, 1992; Burnley and Jones, 2007; Murgatroyd et al., 2011). Consequently, it is the interaction between the trajectory of the $\dot{V}O_2$ slow component, $\dot{V}O_{2max}$, and the capacity to derive energy from substrate-level phosphorylation (PCr hydrolysis, glycolysis leading to the formation of lactate, and the adenylate kinase reaction, Hill’s “oxygen debt”) that dictates the tolerable duration of exercise in the severe-intensity domain (Whipp, 1994; Burnley and Jones, 2007; Murgatroyd et al., 2011). Those interactions, in turn, shape the non-linear relationship between running speed and exercise duration: without the $\dot{V}O_2$ slow component, the critical speed asymptote would be determined by the accumulation of the maximal O₂ deficit, and, as a consequence, occur at the speed associated with $\dot{V}O_{2max}$ (Figure 3C). The non-steady state behaviour of the $\dot{V}O_2$ slow component means that the critical speed asymptote necessarily occurs at a running speed below $\dot{V}O_{2max}$ (Figure 3D).”

4. Lines 426-429. Given the commonly posed argument that women are "less fatigable" than men, could a few sentences be added here about the physiological mechanisms underlying the (absence of) differences in normalized power-duration curves between men and women? I'm not suggesting a detailed description, but a broad muscle energetic context, of the sort provided earlier in the manuscript, to once again frame the male/female performance records in a physiological basis.

This is an interesting and important issue. However, the notion that females are more fatigue resistant than males is a complex and contested issue involving multiple lines of experimental evidence. We respectfully consider that, under the present specific remit of highlighting A.V. Hill’s contributions, that such an analysis would detract from the current context.

5. Lines 449-453. For me, this section is misleading. Reaching a greater $\dot{V}O_2$ more quickly with a fast-start strategy does not necessarily equate to "faster $\dot{V}O_2$ kinetics at exercise onset". In the studies of Sandals et al., 2006, Jones et al., 2008, Bailey et al. 2011 and Wood et al 2014, the faster start causes a greater percentage of $\dot{V}O_{2max}$ to be attained, and/or attained more rapidly, such that a mean response time is reduced. But this experimental method does not account for the energy demand, which is also greater at the beginning of the fast start strategy. Therefore, without knowledge of the energy demand, then it is not possible to infer that the time constant of $\dot{V}O_2$ is actually lower ("faster $\dot{V}O_2$ kinetics at exercise onset") or whether the "account withdrawal" is, in fact, lower (which is the assumption of faster $\dot{V}O_2$

kinetics). The fast start may confer a performance advantage, but these studies do not provide evidence that this advantage is due to "faster VO₂ kinetics".

Thank you, we agree with the premise here – the VO₂ response is changed (i.e., faster absolute dynamics) because VO₂ is projecting towards a greater amplitude rather than exhibiting faster kinetics per se, and we have altered the wording accordingly.

Specific Comments (line)

20 ... A. V. Hill authored three...

Thank you, changed.

29 Importantly or impactfully? My feeling is that "importance" is up to the reader to decide, whereas impact is evident in the historical record.

Agreed, changed.

30 ...characteristics that reflect...

Changed as requested.

32 Fatigue is such a laden term. Without the space in the abstract to develop it, perhaps the following would be a safer statement "...nature of mechanisms limiting exercise tolerance."

Thank you, changed.

44 Perhaps "...British Association for the Advancement of Science..."

Thank you, we have changed the shorter name for the full name here.

50 Perhaps "...of the 1922 Nobel Laureate..."

Thank you, changed.

82 Can you infer, and state what it was, that Hill thought was "extremely interesting and suggestive" about Kennelly's paper?

We are not sure quite what it was, and to do so would be pure speculation, but the paper provided some source data for Hill's own analysis. Although we don't go into it in the manuscript (it is something of a side issue), Kennelly was at pains to point out that the power law was an approximate description of the speed-duration relationship, with data deviating from the fitted line in many cases. This may be why Hill did not rely exclusively on the same mathematical description in his own work. Again, however, we consider this too speculative to examine within the manuscript.

128 Suggest "...the model error associated with events <3 min in..."

Thank you, we have changed the phrasing but retained the "<2 min" part. This is because the data point at just over 7 m/s is an event lasting 114 s, and it is only data points above it that

deviate substantially from the fitted line. We have also taken the liberty of changing “significantly” to “substantially” at the end of the sentence, given the statistical connotations of “significant”.

140 "...in which similar baseline physiological status exists..."

Thank you, changed.

180-185 I can help but think that a reference to a Whipp (et al?) paper is warranted here, but I can't find the paper I am looking for, which shows the O₂ deficit during exercise that asymptotes above VO₂max. Whipp showed that the calculated accumulated O₂ deficit is constant for exercise that VO₂ asymptotes above VO₂max, and can be calculated from VO₂*tau (equation 3.20 in Ch 3. Of Jones and Poole VO₂ kinetics book).

We think we know which figure you mean! We think it is from Whipp, B.J. (1994). The slow component of oxygen uptake kinetics during heavy exercise. *Med Sci Sports Exerc* **26**, 1319-1326. The figure itself is a schematic illustration of the difference between the curvature constant and the O₂ deficit – The O₂ deficit and the area representing energy transfer above VO₂max could not both be constant for the power-duration relationship to be hyperbolic. We have shoe-horned the great man into an existing citation series. The 1994 paper is also cited elsewhere.

195-200 Perhaps include citation here to doi:10.1007/s00421-003-0870-y

Thank you, added.

248 ...Figure 2A...

Added.

252-254 I'm not certain what is being referred to in this sentence. Is it only the <2 min data points or are you also referring to durations at the other end of the speed-duration plot? Up to 15 min or beyond 15 min? Please clarify.

Thank you; reviewer 1 asked us to clarify the wording here too. We are referring only to the short end of the speed-duration plot here.

261. Does Fig 1 in this quote related to Fig 1 in the current manuscript? I found the text is this quote rather abstract. Could some additional explanation be added to describe how this relates to Fig 1 (or 2) in the manuscript?

We appreciate the reviewer's concern, but Hill's Figure 1 and our Figure 1 are the same figure, and so the quote works as stated.

279 Re. "steady state and non-steady state domains". I was struck by the Hill quote on line 161 that used the thermodynamically correct term "dynamic equilibrium". You have an opportunity here to correct the literature on the use of the term "steady-state" in relation to these exercise intensity domains. "Steady-state" describes a condition where all system variables remain constant over time. It requires that at all variables in a system (the body, in this case) remain constant as time changes. There must be no change in mass or energy over

time. This is not true for exercise mildly below CS/CP, where, for example, muscle glycogen is continually falling as exercise continues. Dynamic equilibrium better describes the condition of moderate and heavy intensity exercise, where typically measured variables are stable but this stability is maintained by changes internal to the system (body) that are not typically observed, such as mobilization of fatty acid and/or glucose from one tissue compartment for use in another.

Thank you, we have added a brief section discussing this point as part of the section change related to metabolism above.

286 You might consider including a description of how extreme duration events fit into this scheme e.g., doi:10.1126/sciadv.aaw0341.

We have considered this, and the notion of extreme and distinct limits to exercise performance is interesting. We have mentioned this work in a recent paper in relation to migratory flight in birds (Burnley, 2023), but the events under consideration in the current manuscript play out over seconds, minutes and hours rather than days or weeks. As a result, this fascinating work is difficult to incorporate.

300 ... glycogen utilisation in exercising human muscle would...

Thank you, changed.

302 Why is it reasonable to use an RER of 0.9? This seems low for "racing speeds", even for longer duration races such as the marathon.

We discussed this point in the early drafts of the paper. Initially we calculated the O₂ requirement assuming an RER of 1.0, and then decided on the more conservative 0.9 based on the data of Clark et al. (2019), who reported RERs of ~0.84-0.93 for 2 hours of cycling exercise in the heavy intensity domain. Choosing 1.0 would have reduced the O₂ requirement by 0.1 L.min⁻¹ and the relative VO₂max by 2 mL.kg⁻¹.min⁻¹, but the broader point (that it is a feasible intensity for athletes of the day to attain) remains true.

327 ... from Figure 6 in Hill (1925c)."

Thank you, changed.

334 Suggest to end the sentence at "...compete in ultramarathon events. Ultramarathon runners...". There are likely many reasons for this and there is no advantage to speculation here.

We agree and have removed the clause in question.

397 ...curves provide the...

Thank you, this has been amended.

897 Suggest to include a legend within Figure 6 to explain the different symbols and colours. Is color necessary?

Thank you, the addition of a legend is useful, and it makes the use of colour easier to justify by colour coding the male and female symbols.

Referee #3:

This is a very interesting review of A.V. Hill's 100 years old analysis of world records in some forms of human locomotion, as reported in his three fascinating papers published in 1925. The authors deal mainly with running and, to a lesser extent, with swimming and cycling and clearly show that Hill's outstanding approach to the energetics of human locomotion is essentially supported by 100 years of detailed experimental studies in this and related matters. The authors are to be praised for their profound analysis of Hill's ideas, as well as for their detailed review of 100 years literature in these matters.

We would like to thank the reviewer for this most positive review of our manuscript and extremely generous praise. We are glad you enjoyed reading it and hope that you will be joined by a broad swathe of the JP readership. We have made changes to the text based on the reviewer's comments, and the manuscript has been much improved as a consequence.

A few comments are reported below for the authors' convenience.

General.

The legend to figure 3, lines 847 - 848, reports: "In both plots it is assumed that the primary VO₂ gain is 10 mL.min⁻¹. W⁻¹.....". In fact, this is the efficiency of the transformation of metabolic to mechanical power. Indeed, assuming a Respiratory Quotient of 0.95, the consumption of 1 L O₂ in the human body yields 20.9 kJ; as such a "primary gain" of 10 mL.min⁻¹.W⁻¹ is equivalent to a mechanical efficiency of 0.287. I would therefore suggest the term "mechanical efficiency" rather than "primary gain". Furthermore the efficiency, as estimated from the "steady state" values (for $t \approx 840$ s) of net VO₂ (above resting) and CP of panels B and D of figure 3, assuming an RQ of 0.95, is ≈ 0.21 , i.e. essentially smaller than 0.287.

We thank the reviewer for raising this important point, and we agree with what is written in principle. A primary gain is indeed equivalent to the net mechanical efficiency calculated, but we would caution that in the severe intensity domain there is no steady state, RER would be appreciably above 1.0, and it would be declining during the effort. Thus, the calculated efficiency would also be changing over time. The "primary gain" is a standard measure relating power to VO₂ in the VO₂ kinetics literature, and we have used it here simply to provide a "sensible" VO₂ values after ~ 2 min of exercise. A slow component subsequently drives VO₂ to VO₂max, and when calculated for the longest duration trial the mechanical efficiency has indeed declined from $\sim 29\%$ to $\sim 21\%$ (primary gain increases from 10 to 13 mL.min⁻¹.W⁻¹). The inexorable degradation of mechanical efficiency is a hallmark of the severe intensity domain, as shown in the figure. We respectfully ask for the reviewer's understanding on this issue.

We have added a brief note to the legend of Figure 3 to highlight the reviewer's suggestion.

Again referring to Fig. 3, panels B and D are not coherent with the effects of the acceleration occurring at the beginning of the shortest sprint distances (100 to 400 m). In this case the metabolic power increases abruptly in the first few seconds of the run, and declines in its second part, whereas the mechanical power, after the initial abrupt rise, remains substantially constant, thus leading to a substantial increase of the mechanical efficiency in the second part of the run. This state of affairs is described in detail in some recent studies (two of which are reported below) that have escaped the authors' attention. I do not suggest that the authors go into such details, particularly so because they apply only to the very short distances mentioned above. They may, if they wish, add a very short note on this problem that, in any case does not deserve any detailed discussion.

We thank the reviewer for highlighting this point with regard to the initial phases of athletic races. However, Figure 3 is an illustration of constant load cycling efforts from an unloaded pedalling baseline, and therefore there is no acceleration phase. Moreover, the shortest effort presented in Figure 3 is ~2 min in duration and thus even with an acceleration phase the influence on the energetics of the effort would be small. That said, we agree that these considerations are important for the energetics of short athletic events.

- The energy cost of sprint running and the energy balance of current world records from 100 to 5000 m. P.E. di Prampero, C. Osgnach. In: Biomechanics of Training and Testing. J.-B. Morin, P. Samozino eds. (Springer International), Ch. 12, PP 269 - 297, 2018.

- Mechanical and metabolic power in accelerated running - Part I: the 100-m dash. P.E. di Prampero, C. Osgnach, J-B. Morin, P. Zamparo, G. Pavei. Eur. J. Appl. Physiol. DOI: <https://doi.org/10.1007/s00421-023-05236-x>.

Specific.

- P.7, line173: Jones et al. 2008: please specify whether a, or b;

Thank you, changed.

- P. 11, line 282: Jones et al. 2008: please specify whether a, or b;

Thank you, changed.

- P.17, line 453: : Jones et al. 2008: please specify whether a, or b;

Thank you, changed.

- P.7, line175, Whipp and Ward, 1992: please specify whether a, or b;

Thank you, changed.

- P.11, line405, Whipp and Ward, 1992: please specify whether a, or b;

Thank you, changed.

- P.25, lines 665 and 669: please add a, or b;

Thank you, changed.

- P.25, line 678: the reference to Katz and Katz (1999) is not mentioned in the text;

Thank you, changed.

- P.26, line 704: the reference to Mitchell et al. (2018) is not mentioned in the text;

Thank you, changed.

- P.29, lines 793 and 796: please add a, or b.

Thank you, changed.

Dear Dr Burnley,

Re: JP-TR-2025-288130R1 "Blue Plaque Review Series: A.V. Hill, athletic records, and the birth of exercise physiology" by Mark Burnley, Anni Vanhatalo, David C Poole, and Andrew M Jones

Thank you for submitting your manuscript to The Journal of Physiology. It has been assessed by a Reviewing Editor and by 3 expert referees and we are pleased to tell you that it is acceptable for publication following satisfactory minor revision.

ABSTRACT FIGURES: Authors may use The Journal's premium BioRender account to create/redraw their Abstract Figures (and any other suitable schematic figure). Information on how to access this account is here: <https://physoc.onlinelibrary.wiley.com/journal/14697793/biorender-access>.

REVISION CHECKLIST: Upload a full Response to Referees file. To create your 'Response to Referees' copy all the reports, including any comments from the Senior and Reviewing Editors, into a Microsoft Word, or similar, file and respond to each point, using font or background colour to distinguish comments and responses and upload as the required file type.

We look forward to receiving your revised submission.

Yours sincerely,

Laura Bennet
Senior Editor

EDITOR COMMENTS

Reviewing Editor:

All 3 very experienced referees are satisfied with the revision and congratulate the authors for an excellent review. I concur with their enthusiasm. To use the words of one of the Reviewers: "Provides great insight into the historical development of modern day concepts. Hopefully, beginning researchers will gain an appreciation of prior "old" research, and more experienced researchers will be reminded to consult the "old" research as well."

Only Reviewer 2 has a minor point to make, which the authors should take into consideration.

REFEREE COMMENTS

Referee #1:

Thank you for the conscientious revisions - nice job!

Referee #2:

I thank the authors for their thoughtful consideration of my comments and suggestions, and congratulate them on a very nice article.

I have only one minor suggestion remaining re. lines 337-340 (redlined version). I see now that the authors are using an RER of 0.9 because the calculation relates to a sub-maximal effort ("Elite athletes of the period were likely to sustain such demands comfortably" in the following sentence). Most of the rest of the article is dealing with athletic records, which, by definition, are maximal efforts. That was why I was surprised to see the RER of 0.9 used. Could the authors add a qualifier here to help bring the reader's attention to the use of RER=0.9 being appropriate for a sub-maximal effort? Perhaps: "Nevertheless, the utilization of 300 g of glycogen per hour implies that, at an RER of 0.9 for a sustained effort at approximately 65-70% VO₂max, the athlete would need to sustain a..."

Referee #3:

The authors have appropriately addressed all my comments and suggestions.

END OF COMMENTS

R2 Response to reviewer's comments

We would like to take the opportunity once again to thank the reviewer for reviewing the manuscript and our comments. We have made a minor edit based on those comments and we explain the change below.

Reviewer 2:

I have only one minor suggestion remaining re. lines 337-340 (redlined version). I see now that the authors are using an RER of 0.9 because the calculation relates to a sub-maximal effort ("Elite athletes of the period were likely to sustain such demands comfortably" in the following sentence). Most of the rest of the article is dealing with athletic records, which, by definition, are maximal efforts. That was why I was surprised to see the RER of 0.9 used. Could the authors add a qualifier here to help bring the reader's attention to the use of RER=0.9 being appropriate for a sub-maximal effort? Perhaps: "Nevertheless, the utilization of 300 g of glycogen per hour implies that, at an RER of 0.9 for a sustained effort at approximately 65-70% $\dot{V}O_{2max}$, the athlete would need to sustain a..."

Thank you. We agree that there seems to be a contradiction between the world record analysis and this example of running at a sustained pace for an hour. However, a prolonged maximal "effort" is necessarily performed at a submaximal $\dot{V}O_2$ (and relatively low RER), with maximal effort occurring at the end of the run. For this reason, we have adjusted the sentence in question thus:

"Nevertheless, the utilisation of 300 g of glycogen per hour implies that, at an RER of 0.9, the athlete would need to sustain a submaximal $\dot{V}O_2$ of $\sim 3.9 \text{ L}\cdot\text{min}^{-1}$ ($\sim 56 \text{ mL}\cdot\text{kg}^{-1}\cdot\text{min}^{-1}$, assuming a body mass of 70 kg, an energy density of CHO of $16 \text{ kJ}\cdot\text{g}^{-1}$ and O_2 consumption providing $20.7 \text{ kJ}\cdot\text{L}^{-1}$). Elite athletes of the period were likely to sustain such demands comfortably (e.g., Paavo Nurmi)."

We have added "submaximal" before " $\dot{V}O_2$ " on the second line to emphasise the submaximal nature of the task. We appreciate what the reviewer was suggesting with the use of phrase "for a sustained effort at approximately 65-70% $\dot{V}O_{2max}$ ", but this is likely to underestimate the fractional utilisation of $\dot{V}O_{2max}$ for an athlete. Thus, we suggest that the inclusion of the single word "submaximal" in the original sentence resolves the reviewer's valid original point without making any additional assumptions.

Dear Dr Burnley,

Re: JP-TR-2025-288130R2 "Blue Plaque Review Series: A.V. Hill, athletic records, and the birth of exercise physiology" by Mark Burnley, Anni Vanhatalo, David C Poole, and Andrew M Jones

We are pleased to tell you that your paper has been accepted for publication in The Journal of Physiology.

Authors should note that it is too late at this point to offer corrections prior to proofing. Major corrections at proof stage, such as changes to figures, will be referred to the Editors for approval before they can be incorporated. Only minor changes, such as to style and consistency, should be made at proof stage. Changes that need to be made after proof stage will usually require a formal correction notice.

Yours sincerely,

Laura Bennet
Senior Editor
The Journal of Physiology

P.S. - You can help your research get the attention it deserves! Check out Wiley's free Promotion Guide for best-practice recommendations for promoting your work at www.wileyauthors.com/eeo/guide. You can learn more about Wiley Editing Services which offers professional video, design, and writing services to create shareable video abstracts, infographics, conference posters, lay summaries, and research news stories for your research at www.wileyauthors.com/eeo/promotion.

IMPORTANT NOTICE ABOUT OPEN ACCESS: To assist authors whose funding agencies mandate public access to published research findings sooner than 12 months after publication, The Journal of Physiology allows authors to pay an Open Access (OA) fee to have their papers made freely available immediately on publication.

You can check if your funder or institution has a Wiley Open Access Account here: <https://authorservices.wiley.com/author-resources/Journal-Authors/licensing-and-open-access/open-access/author-compliance-tool.html>.

EDITOR COMMENTS

Reviewing Editor:

I am satisfied with the authors' response. This is an excellent review.